# Hospitalizations and Treatment Outcomes in Patients with Urogenital Tuberculosis in Tashkent, Uzbekistan, 2016–2018

**DOI:** 10.3390/ijerph18094817

**Published:** 2021-04-30

**Authors:** Bakhtiyor Ismatov, Yuliia Sereda, Serine Sahakyan, Jamshid Gadoev, Nargiza Parpieva

**Affiliations:** 1Republican Specialized Scientific and Practical Medical Center of Tuberculosis and Pulmonology, Tashkent 100086, Uzbekistan; nargizaparpieva@gmail.com; 2Independent Consultant, 01116 Kyiv, Ukraine; yulia.v.sereda@gmail.com; 3Armenia and Tuberculosis Research and Prevention Center NGO, Yerevan 0034, Armenia; s.sahakyan@aua.am; 4World Health Organization (WHO) Country Office in Uzbekistan, Tashkent 100100, Uzbekistan; gadoevj@who.int

**Keywords:** length of stay, inpatient care, extrapulmonary tuberculosis, domiciliary care, Central Asia, SORT-IT

## Abstract

Despite the global shift to ambulatory tuberculosis (TB) care, hospitalizations remain common in Uzbekistan. This study examined the duration and determinants of hospitalizations among adult patients (≥18 years) with urogenital TB (UGTB) treated with first-line anti-TB drugs during 2016–2018 in Tashkent, Uzbekistan. This was a cohort study based on the analysis of health records. Of 142 included patients, 77 (54%) were males, the mean (±standard deviation) age was 40 ± 16 years, and 68 (48%) were laboratory-confirmed. A total of 136 (96%) patients were hospitalized during the intensive phase, and 12 (8%) had hospital admissions during the continuation phase of treatment. The median length of stay (LOS) during treatment was 56 days (Interquartile range: 56–58 days). LOS was associated with history of migration (adjusted incidence rate ratio (aIRR): 0.46, 95% confidence interval (CI): 0.32–0.69, *p* < 0.001); UGTB-related surgery (aIRR: 1.18, 95% CI: 1.01–1.38, *p* = 0.045); and hepatitis B comorbidity (aIRR: 3.18, 95% CI: 1.98–5.39, *p* < 0.001). The treatment success was 94% and it was not associated with the LOS. Hospitalization was almost universal among patients with UGTB in Uzbekistan. Future research should focus on finding out what proportion of hospitalizations were not clinically justified and could have been avoided.

## 1. Introduction

Tuberculosis (TB), one of the leading causes of death globally, is mostly represented by pulmonary TB [1]. Extrapulmonary tuberculosis (EPTB) has traditionally received less priority and attention probably due to its non-infectious nature. In 2019, EPTB represented 16% of the 7.1 million patients with TB that were reported to the World Health Organization (WHO) [1]. Urogenital tuberculosis (UGTB) is a form of EPTB related to infectious inflammation of urogenital system organs in any combination, caused by *Mycobacterium tuberculosis* or *Mycobacterium bovis*. Globally, UGTB accounts for 30 to 40% of patients with EPTB [2]. 

EPTB diagnosis, including the urogenital form, is a challenge due to the pauci-bacillary nature of the disease and the non-uniform distribution of microorganisms in the body [3]. Extra-pulmonary specimens may need a decontamination procedure during the sample preparation, which in its turn reduces the sensitivity of culture methods, a gold standard in TB diagnosis [4]. Only one-third of patients with UGTB have X-ray abnormalities and classical symptoms, such as fever, night sweats, and weight loss, are not common [5]. Thus, the diagnosis is often presumptive based on clinical or radiological findings without laboratory confirmation. Moreover, the definition of a satisfactory response to treatment in EPTB is not well-defined and varies across countries [6].

On average, patients with EPTB have longer hospitalizations compared to those with pulmonary TB [7]. Moreover, children, homeless patients, patients with psychiatric or substance abuse issues, and patients with comorbidities and complications tend to have a longer length of stay [8,9,10,11]. Generally, in recent years, there has been an intention to reduce hospitalizations during TB treatment. Several studies showed that treatment models with hospitalization for the full length of the intensive phase and models with ambulatory treatment since the day of diagnosis have similar treatment success [12,13,14]. WHO suggests hospital admissions only for complicated patients with TB, such as with respiratory failure or requiring surgery, patients with severe forms of the disease, and life-threatening or serious adverse drug events [15]. Hospitalization is also considered when effective and safe treatment cannot be ensured in an outpatient setting [15].

Uzbekistan is among the 25 countries with the largest proportion of EPTB, accounting for 35% of 16,272 new and relapsed patients with TB in 2019 [1]. However, the burden of UGTB is not known. Since 2014, it has been recommended that drug-susceptible TB (DSTB) patients in Uzbekistan receive their intensive phase of the treatment in an outpatient TB facility. However, this is not frequently followed because of the continued incentivization in favor of hospital-based care (such as financing of hospitals based on bed numbers and occupancy rates), and the relatively underdeveloped and underfinanced ambulatory sector [16].

The long-term inpatient care puts an additional burden on the limited healthcare resources. Previous research showed that hospitalization may account for half of all TB treatment costs [17]. Moreover, hospitalization increases the risk of TB nosocomial transmission to healthcare workers and other patients and may lead to mental health complications due to isolation [18,19]. None of the previous studies explored the length of hospitalization during TB treatment in Uzbekistan or Central Asia. We aimed to determine the duration of hospitalization during the intensive and continuation phases of treatment, its associated factors, and its relation to successful treatment outcomes among patients with UGTB in a tertiary care hospital in Uzbekistan.

## 2. Materials and Methods

### 2.1. Study Design

This was a cohort study based on the secondary analysis of patients’ records.

### 2.2. General Setting

Uzbekistan is a lower middle-income country in Central Asia with a population of 33 million, two-thirds of whom live in rural areas [20] The country is divided into 12 provinces, an autonomous republic (Karakalpakstan), and a capital city (Tashkent).

### 2.3. Specific Setting

In Uzbekistan, TB services are vertically structured and provided at central, oblast, district, and primary health care levels. Funding for the National TB program comes mainly from external donors, in particular, The Global Fund to Fight AIDS, Tuberculosis and Malaria (Global Fund), The United States Agency for International Development (USAID), and Médecins Sans Frontières (MSF).

The study was conducted in the Republican Specialized Scientific Research Medical Center of Phthisiology and Pulmonology (RSSRMCPP), a tertiary referral center with branches in regions (district-level TB centers). The majority of patients are referred to RSSPMCTP from the secondary care (regional TB hospitals) and other tertiary care facilities (infectious diseases clinics, urology centers, etc.). The UGTB department at RSSPMCTP consists of seven physicians and 12 nurses and provides diagnostic and treatment services including hospitalization and surgery, as required. In total, 55 hospital beds are available for patients with UGTB at RSSRMCPP.

In Uzbekistan, UGTB diagnosis is usually classified into three categories: urinary tract tuberculosis, genital tuberculosis, both urinary tract and genital tuberculosis. Kidney tuberculosis is considered as urinary tract tuberculosis. UGTB diagnosis and treatment regimens follow the WHO guidelines [21]. Culture, histopathology, intravenous pyelography, laparoscopy, cystoscopy, and biopsy are the methods employed in the UGTB diagnosis. Clinical diagnosis for drug-susceptible UGTB is based on the patient’s history and no history of drug-resistant TB (DR-TB) among the patient’s contacts. Treatment for drug-susceptible UGTB consists of intensive and continuation phases. The intensive phase consists of treatment with four drugs (isoniazid, rifampicin, pyrazinamide, and ethambutol) and lasts 56 days. Patients with UGTB are usually hospitalized for the full duration of the intensive phase. As per the national guidelines, intensive-phase treatment can be extended up to 84 days among patients with a positive culture and those with urinary biomarkers of inflammation or treatment complications. The continuation phase consists of two drugs (rifampicin and isoniazid) and is provided for six months in an outpatient setting at RSSRMCPP, regional TB centers, or primary care facilities. The continuation phase is extended up to seven months for patients with disseminated TB, TB/human immunodeficiency virus (HIV) and when urinary tract destruction without bacterial excretion persists for over six months. The continuation phase of treatment can last for 8–12 months, if there is coexisting tuberculous meningitis. Patients with UGTB are hospitalized during the continuation phase in cases of serious adverse effects or need for surgery. Definitions of UGTB treatment outcomes in Uzbekistan are presented in Appendix A. Decisions regarding the length of treatment and duration of hospitalization for each patient with UGTB is made by the “consilium”, a committee of physicians authorized to make treatment decisions.

### 2.4. Study Population and Period

We included all patients who met the following criteria: age ≥18 years, diagnosed clinically with presumed or bacteriologically confirmed drug-susceptible UGTB, and received first-line treatment during 2016–2018 in the UGTB department at the RSSPMCTP in Tashkent, Uzbekistan. We excluded patients with a previous history of UGTB and patients with drug-resistant UGTB as they have a different treatment regimen and longer treatment duration, which requires a separate analysis.

### 2.5. Variables, Definitions, Data Sources

The variables related to the study objectives were extracted from paper-based health records at the RSSPMCTP. These were sociodemographic, behavioral, and clinical characteristics of patients at admission; dates of diagnosis, treatment initiation, hospitalizations and discharges; count and period of UGTB-related surgeries; and presence of serious adverse drug events during treatment. Serious adverse events are events that are (a) life-threatening; (b) lead to disability or permanent damage; (c) require hospitalization or its extension; (d) lead to a congenital anomaly or birth defect; (e) other events that may jeopardize the patient [22]. Data on final treatment outcomes was requested from regional TB centers as most patients with UGTB who completed intensive-phase in RSSPMCTP, received continuation-phase treatment in their residential area.

Primary outcomes were (i) total length of stay, defined as the duration of all hospitalization episodes during UGTB treatment; (ii) length of stay during the intensive phase of treatment; and (iii) length of stay during the continuation phase of treatment. Secondary outcomes were (i) “extended intensive-phase hospitalization” defined as hospitalization for the full length of the extended intensive phase (84 days or longer), and (ii) presence of hospitalization(s) with a duration of one day or longer during the continuation phase of treatment. Final treatment outcomes were classified into favorable (cured and treatment completed) and unfavorable outcomes (death, lost to follow-up, failure).

### 2.6. Data Entry and Analysis

Data were collected during May–October 2020. We entered selected variables from paper health records into a structured EpiData database created for the purpose (version 3.1 for entry EpiData Association, Odense, Denmark). Fifteen percent of records were double-entered and validated.

Characteristics of the study participants were summarized with descriptive statistics using frequencies and proportions (for categorical variables) and mean and standard deviation (SD) or median and interquartile range (IQR) for continuous variables, as appropriate. We calculated proportions of patients hospitalized during the intensive and continuation phases of treatment. The total lengths of stays during intensive and continuation phases of treatment were summarized with medians and interquartile ranges for each phase of treatment. Considering the count nature of primary outcome and presence of overdispersion, negative binomial regressions were used to assess factors associated with total length of stay. Variables for the adjusted model were selected based on AIC (Akaike Information Criteria) stepwise testing. Age and sex were included in the adjusted model regardless of the results of stepwise testing. We calculated proportions of unfavorable treatment outcomes with 95% confidence intervals among patients with different patterns of hospitalization, such as (i) outpatient treatment only; (ii) standard intensive phase hospitalization and no hospitalizations during continuation phase: the shortest length of stay; (iii) extended intensive phase hospitalization and no hospitalizations during continuation phase; and (iv) extended intensive phase hospitalization and hospitalizations during continuation phase: the longest length of stay. A level of significance was set at α = 0.05 for all analyses. Analysis was conducted using R software, version 3.5.2 (Copyright (C) 2018 The R Foundation for Statistical Computing).

## 3. Results

### 3.1. Patient Characteristics

A total of 142 patients were included in the study. The mean (± standard deviation) age was 40 ± 16 years and about half (n = 77, 54%) were males (Table 1). The majority of patients were from rural areas (n = 113, 80%). All patients had satisfactory living conditions by judgment of their TB physicians. Only three patients (2%) had a history of alcohol abuse at the time of admission while tobacco smoking was more common (n = 30, 21%).

Nine patients (6%) had a previous history of TB, pulmonary or extrapulmonary, but not except UGTB. About half of the patients (n = 68, 48%) were laboratory confirmed and the rest were clinically diagnosed (n = 74, 52%). Overall, the diagnosis was confirmed by microscopy in 9 patients (6%), by Xpert MTB/Rif in 56 patients (39%), and by culture in 30 patients (21%). A total of 90 patients (66%) had comorbid conditions, and the most common conditions were non-specific urinary tract infection (n = 63, 44%), hypertension (n = 33, 23%), and anemia (n = 33, 23%).

### 3.2. Hospitalizations and Length of Stay

All 142 patients were hospitalized during the stage of diagnosis. On average, patients spent a week (median: 7 days, interquartile range (IQR): 5–8) in the UGTB inpatient department before confirmation of diagnosis and initiation of treatment.

Most patients (96%, n = 136) were hospitalized during the intensive phase of treatment. Of them, 43% (59/136) had an extended intensive phase. The total length of stay during the intensive phase of treatment ranged from 23 to 113 days with a median of 56 days (IQR: 56–57). Six patients were treated in the outpatient setting; they received medication at home and were supervised by nurses. Of the six patients who had outpatient treatment since the day of diagnosis, four patients had an extended intensive phase of treatment.

During the continuation phase of treatment, 12 of 142 (8%) patients had hospitalizations, and length of stay ranged from 7 to 275 days with a median of 18.5 days (IQR: 12–30). Of those 12 patients, 10 were admitted once during the continuation phase of treatment, one patient was admitted twice, and one patient was admitted three times.

Overall, 73 of 142 (51%) patients were hospitalized only during the intensive phase of treatment and stayed in the hospital for 56 days, the full length of the standard intensive-phase. Fifty-one patients (36%) had an extended intensive phase and stayed in the hospital over 56 days, but they did not have any hospitalizations during the continuation phase. Eight patients (6%) had the longest length of stay with extended intensive phase hospitalization and admission(s) to the hospital during the continuation phase. The total length of stay during intensive and continuation phases of treatment ranged from zero to 360 days with a median of 56 days (IQR: 56–58 days).

### 3.3. Factors Associated with Length of Stay

History of migration was associated with shorter length of stay during UGTB treatment (adjusted incidence rate ratio (aIRR): 0.46, 95% confidence interval (CI): 0.32–0.69, *p* < 0.001) while surgery and hepatitis B were linked to longer length of stay (aIRR: 1.18, 95% CI: 1.01–1.38, *p* = 0.045 1.18 and aIRR: 3.18, 95% CI: 1.98–5.39, *p* < 0.001, respectively) (Table 2). Alcohol abuse was a potential predictor of longer length of stay, the association was marginally insignificant (aIRR: 2.01, 95% CI: 0.99–4.21, *p* = 0.07).

### 3.4. Treatment Outcomes

Overall, 94% (134/142) of patients with UGTB achieved treatment success. One patient (1%) died after 2.5 months of treatment. Seven patients (5%) had treatment failure. Proportion of unfavorable treatment outcomes was comparable among patients with standard and extended intensive-phase hospitalization (2/73, 3% and 1/51, 2%, respectively, *p* = 0.71) (Figure 1).

## 4. Discussion

This study explored duration of hospitalization among patients with drug-susceptible UGTB in a tertiary care center in Uzbekistan. We found high rates of hospitalization with nearly everyone hospitalized during the intensive phase of treatment, and nearly one in ten patients had repeated hospital admission during the continuation phase of treatment. This is not in line with the WHO recommendations and national treatment guidelines in Uzbekistan that encourage ambulatory TB treatment [15].

In our study, the median length of stay was 56 days for patients with UGTB, which is similar to the regional average for the duration of hospitalization among patients with drug-susceptible TB in Eastern Europe and Central Asia [1]. In countries with a predominantly ambulatory model of TB treatment, such as the United States or Italy, the average length of stay for extrapulmonary TB has been reported to be between 13–22 days [7,23]. Over-hospitalization of patients with TB that is not always justified by clinical need has been described previously in health care systems similar to Uzbekistan, which inherited the Soviet model of hospital-based management [16,24]. Previous qualitative research in the country showed that structural barriers, such as the hospital financing mechanism in the country, which is based on occupancy rates, and lack of comprehensive outpatient care preclude the scale-up of ambulatory TB treatment [16].

In the study population, the admissions to RSSPMCTP began on average a week before UGTB treatment initiation. As UGTB diagnosis is often neglected, patients encounter TB care with severe disease and need inpatient symptomatic treatment before UGTB diagnosis confirmation [25]. Considering the length of stay during treatment and duration of hospitalization before treatment initiation, first-line patients with UGTB had more than two months of inpatient treatment on average.

In our study, two-thirds of patients with UGTB had comorbidities, and 6% of them experienced serious adverse events during the treatment, such as life-threatening events or events requiring hospitalizations. This finding may indicate that comorbidities contributed to a more severe condition that required hospitalization [26]. In the previous research, patients with TB and comorbidities, particularly renal and liver diseases, diabetes mellitus, HIV, or cancer, were more likely to be hospitalized and have longer inpatient treatment [8,26,27,28]. Similarly, we found that patients with UGTB and hepatitis B were more likely to stay longer in the hospital. Associations between the length of stay and other comorbidities were not significant.

Hepatitis B and underlying chronic liver disease are well-known risk factors of hepatotoxicity induced by anti-tuberculosis treatment and poor outcomes [29,30]. In our sample, all four participants with hepatitis B developed hepatotoxicity during the intensive phase of TB treatment, and three of them had prolonged initial hospitalization with about three months duration. Unfavorable treatment outcome with treatment failure was observed in one of the four patients with TB and hepatitis B, who had extended intensive phase hospitalization and repeated hospitalizations during the continuation phase.

Our data showed that the patients with UGTB who underwent surgery during their treatment were more likely to have a longer length of stay, which is quite expected. In UGTB clinical practice, the intensive phase of treatment starts with the standardized anti-TB treatment; if the patient’s health is not improved during the first month of the intensive phase of treatment, a surgical intervention is considered [31]. Common kinds of surgery in patients with UGTB in RSSPMCTP were nephroureterectomy, nephrectomy, urinary diversion, orchiectomy, ureteral stent placement, and reconstructive urethral surgery. Patients with UGTB who had surgery usually stayed in the hospital for an extra month after 56 days of initial hospitalization (standard intensive phase).

The history of labor migration was associated with shorter length of stay in our study. Migrants are a key affected population with respect to TB in Central Asia, including Uzbekistan, due to low socio-economic status, limited access to health care while working abroad, and increased vulnerability to HIV infection [32,33]. Considering that prolonged hospitalization leads to loss of productivity and income, migrants tend to avoid hospitalization regardless of the actual needs [34].

High rates of hospitalization and long length of stay might be influenced also by the low socio-economic status of patients with TB. Unemployment, poverty, and homelessness are considered by health workers as social reasons to justify hospital admissions [35]. Some patients may prefer to stay in the hospital to reduce expenditure, such as nutrition or travel expenses for medication refills and treatment monitoring [36]. Previous studies in Uzbekistan have shown that one in four patients with TB is unemployed and at increased risk of loss to follow up [37,38].

Repeated hospital admissions were not common in our study population, which is consistent with previous research. Patients usually start TB treatment in-hospital and initial hospitalization contributes the most to the total length of stay during treatment [8,28]. Research in the United States and Canada, countries that prioritize reduction of hospital admissions, showed similar proportions of patients with TB with multiple hospitalization episodes during treatment (7 and 8%, respectively) [8,39].

The final treatment outcomes were excellent—the treatment success was 94% and there were no patients lost to follow-up. The treatment success reported in the study is comparable with an overall estimate of the TB treatment success rate in Uzbekistan among new patients and relapses (92% in 2018) and with the international literature suggesting a high success rate in UGTB treatment [31].

There were some limitations in our study. While the initial aim was to include all patients who met inclusion criteria in the study, it was not possible to collect data on 302 of 444 (68%) of patients with UGTB. The data collection period overlapped with the COVID-19 pandemic and TB centers were involved in the pandemic response, which made data collection challenging. Hence, the study was subject to a non-response bias. Our study population included patients with UGTB diagnosed at a tertiary care hospital and may not be representative of all patients with UGTB in the country. Variables, such as alcohol abuse and tobacco smoking at admission were not assessed by standardized screening tools and so were likely underestimated. Patient records had limited information on social determinants of health. Health records included information on living conditions but data on socio-economic status, homelessness, and substance abuse were not available. These variables are important predictors of hospital admissions and length of stay as per the published literature [40]. Previous research has shown that hospital-acquired infections, particularly infections caused by drug-resistant microorganisms, are common in patients with TB, which prolong the length of stay [41,42]. Study participants were not routinely screened for hospital-acquired infections and neither were these infections monitored at RSSPMCTP. Therefore, we were not able to assess the impact of hospital-acquired infections on the length of stay. One limitation was the small sample size that restricted the analysis of factors associated with the length of stay. The finding on the link between hepatitis B and longer length of stay did not have adequate statistical power given only four patients in the sample had hepatitis B.

Despite these limitations, there are two important programmatic implications. First, the National TB program needs a better understanding of what proportion of hospitalizations and what length of stay in patients with TB are justified by clinical needs. One study in the United States showed that up to 40% of hospitalizations in patients with TB were avoidable [19]. This study considered both clinical and social criteria for hospitalization: clinically unstable on admission; admitted to the intensive care unit or intubated; receiving home or nursing home care; homeless or living in a congregate setting; recent alcohol/drug abuser; aged < 5 years; admitting diagnosis of a severe form of TB; and presence of mental illness [19]. A study in the Russian Federation found that about 20% of hospital admissions in patients with TB would not be justified when clinical (severity of disease), public health (risk of transmission), social (risk behaviors and socio-economic status), and health-system factors (access to outpatient care) are considered as hospitalization criteria [35]. Similar research, particularly in patients with UGTB, will help the National Tuberculosis Programme in Uzbekistan to infer hospitalization data more appropriately. Second, high admission rates at the tertiary care found in the study and multiple comorbidities among patients with UGTB may indicate a delay in diagnosis that contributes to a severe illness and prolonged hospitalization [25]. In this regard, there is a need to understand what contributes most to the delay (late presentation by the patients to the health system, low clinical suspicion of UGTB on the part of health care providers at primary care, non-productive referrals within the health system) and address these barriers.

## 5. Conclusions

Our study demonstrated that the hospitalization rates among patients with UGTB in Uzbekistan were quite high, despite WHO and national recommendations in favor of a decentralized, ambulatory model of care. We found that the factors related to the longer length of stay include hepatitis B and surgery while history of labor migration was associated with shorter length of stay. Future research should focus on finding out what proportion of hospitalizations were not clinically justified and could have been avoided.

## Figures and Tables

**Figure 1 ijerph-18-04817-f001:**
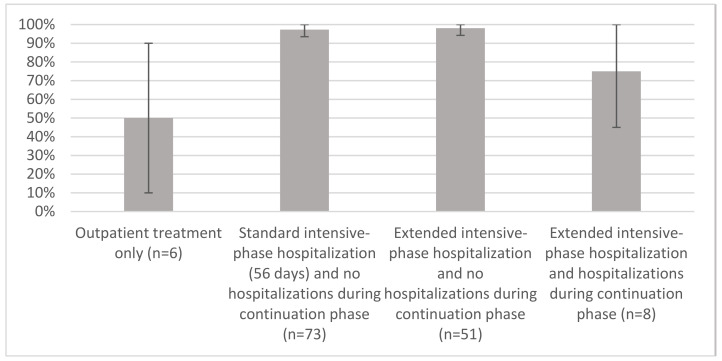
Proportions of drug-susceptible patients who were cured or completed UGTB treatment by different hospitalization patterns, Tashkent, Uzbekistan, 2016–2018, % and 95% confidence intervals.

**Table 1 ijerph-18-04817-t001:** Demographic, clinical, and diagnostic profile of patients with urogenital tuberculosis (UGTB) who received first-line treatment in Tashkent, Uzbekistan, from 2016–2018.

Characteristics/Variables		N	(%)
	Total	142	(100)
Age, mean (SD) categories	Mean (SD, min–max)	40 (16, 18–80)	
	18–34	67	(47)
	35–54	45	(32)
	55–80	30	(21)
Sex	Male	77	(54)
	Female	65	(46)
Type of residence	Urban	29	(20)
	Rural	113	(80)
Alcohol abuse	Yes	3	(2)
	No/Not recorded	139	(98)
Current tobacco smoking	Yes	30	(21)
	No/Not recorded	112	(79)
Labor migration to other countries in the past six months	Yes	7	(5)
	No/Not recorded	135	(95)
Bacteriological confirmation	Clinically diagnosed	74	(52)
	Laboratory confirmed	68	(48)
Urine microscopy at admission	M. *tb* detected	9	(6)
	M. *tb* not detected	133	(94)
Xpert MTB/RIF at admission	Rifampicin sensitive	56	(39)
	M. *tb* not detected	84	(59)
	Not recorded	2	(2)
Culture at admission	M. *tb* detected	30	(21)
	M. *tb* not detected	111	(78)
	Contamination	1	(1)
Disseminated TB	No	118	(83)
	Yes	23	(16)
	Not recorded	1	(1)
Type of UGTB	Urinary tract TB	87	(61)
	Genital TB	14	(10)
	Both	41	(29)
Previous history of TB	Pulmonary	6	(4)
	Extrapulmonary (except UGTB)	3	(2)
	None	131	(92)
	Not recorded	2	(1)
BMI at admission	<18.5	15	(11)
	18.5–24.9	118	(83)
	25.0–29.9	6	(4)
	30.0–39.9	3	(2)
Comorbidities	Hypertension	33	(23)
	Diabetes mellitus	11	(8)
	Anemia	33	(23)
	Hepatitis B	4	(3)
	Hepatitis C	2	(1)
	HIV status	3	(2)
	Non-specific urinary tract infection	63	(44)
	Other ^1^	43	(30)
	Any comorbidity	90	(66)
Surgery for UGTB	None	93	(65)
	During intensive phase	45	(32)
	During continuation phase	2	(1)
	During both phases	2	(1)
Serious adverse events during treatment	Reported	8	(6)
	Not reported	134	(94)

^1^ Other comorbidities were urogenital disorders (n = 31), cardiovascular diseases (n = 13), neurological diseases (n = 3), cancer (n = 2).

**Table 2 ijerph-18-04817-t002:** Factors associated with the total length of stay among patients with UGTB who received first-line treatment, Tashkent, Uzbekistan, 2016–2018 (N = 142).

	Median LOS	Unadjusted		Adjusted	
	(Q1; Q2)	IRR (95% CI)	*p*	IRR (95% CI)	*p*
Age groups, years					
18–34	56 (56; 58)	ref.		ref.	
35–54	56 (56; 57)	1.05 (0.86; 1.28)	0.63	0.92 (0.77; 1.09)	0.31
55–80	57 (55; 70)	0.94 (0.75; 1.18)	0.56	0.93 (0.74; 1.17)	0.56
Sex					
Male	56 (56; 57)	ref.		ref.	
Female	56 (56; 58)	0.95 (0.80; 1.13)	0.55	0.95 (0.82; 1.1)	0.49
Type of residence					
Urban	57 (55; 70)	ref.		-	-
Rural	56 (56; 57)	1.00 (0.81; 1.24)	0.99	-	-
Alcohol abuse at admission					
No	56 (56; 58)	ref.		ref.	
Yes	56 (56; 105)	1.16 (0.67; 2.23)	0.62	2.01 (0.99; 4.21)	0.06
Current tobacco smoking at admission					
No	56 (56; 58)	ref.		-	-
Yes	56 (55; 57)	0.97 (0.79; 1.20)	0.78	-	-
Labor migration to other countries in the past 6 months prior to admission					
No	56 (56; 58)	ref.		ref.	
Yes	56 (0; 56)	0.63 (0.43; 0.95)	0.02	0.46 (0.32; 0.69)	<0.001
UGTB diagnosis					
Clinically diagnosed	56 (56; 58)	ref.		ref.	
Laboratory confirmed	56 (55; 59)	1.08 (0.91; 1.28)	0.40	1.07 (0.92; 1.25)	0.34
Disseminated TB					
No	56 (56; 57)	ref.		-	-
Yes	56 (55; 76)	1.00 (0.79; 1.27)	0.98	-	-
Type of UGTB					
Urinary tract and genital TB	56 (56; 58)	ref.			
Urinary tract TB only	56 (56; 58)	1.02 (0.84; 1.24)	0.82	-	-
Genital TB only	56 (55; 57)	0.85 (0.63; 1.18)	0.33	-	-
Previous history of pulmonary or extrapulmonary TB, except UGTB					
No	56 (56; 57)	ref.		-	-
Yes	70 (56; 84)	1.13 (0.81; 1.65)	0.49	-	-
BMI					
<18.5	57 (56; 67)	ref.		-	-
18.5–24.9	56 (56; 58)	1.08 (0.8; 1.41)	0.61	-	-
>24.9	56 (55; 56)	0.87 (0.57; 1.36)	0.54	-	-
Hypertension					
No	56 (56; 58)	ref.		ref.	
Yes	57 (55; 67)	1.10 (0.90; 1.36)	0.34	1.12 (0.91; 1.4)	0.29
Diabetes Mellitus					
No	56 (56; 58)	ref.		ref.	
Yes	55 (54; 57)	0.86 (0.63; 1.20)	0.35	0.79 (0.57; 1.09)	0.15
Anemia					
No	56 (56; 58)	ref.		-	-
Yes	56 (55; 57)	0.97 (0.80; 1.20)	0.80	-	-
Hepatitis B					
No	56 (56; 57)	ref.		ref.	
Yes	125 (85; 248)	2.80 (1.82; 4.57)	<0.001	3.18 (1.98; 5.39)	<0.001
Hepatitis C (ref. no)					
No	56 (56; 58)	ref.		-	-
Yes	81 (56; 105)	1.29 (0.67; 2.92)	0.49	-	-
HIV status					
Negative	56 (56; 58)	ref.		ref.	
Positive	105 (56; 113)	1.48 (0.86; 2.82)	0.19	0.53 (0.25; 1.14)	0.12
Non-specific urinary tract infection					
No	56 (56; 58)	ref.		-	-
Yes	56 (55; 58)	1.02 (0.86; 1.22)	0.80	-	-
Surgery during treatment					
No	56 (55; 57)	ref.		ref.	
Yes	57 (56; 85)	1.15 (0.96; 1.37)	0.14	1.18 (1.01; 1.38)	0.045
Serious adverse events during treatment					
No	56 (56; 58)	ref.		-	-
Yes	56 (55; 71)	1.02 (0.72; 1.51)	0.91	-	-

## Data Availability

The data that support the findings of this study are available from the corresponding author, B.I., upon reasonable request.

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
