# Peer review of "Hospitalizations and Treatment Outcomes in Patients with Urogenital Tuberculosis in Tashkent, Uzbekistan, 2016–2018"

_ijerph, 2021, doi:10.3390/ijerph18094817_

Round 1

Reviewer 1 Report

This is an important manuscript addressing hospitalization and LOS of TB patients in Uzbekistan. Overall, the paper is well written, and care was taken to address the various limitation of an observational study based on medical record review.  My comments are mostly minor.

1) Perhaps longer LOS is partly driven by hospital acquired infections? Can some data be provided to discount this concern?

2) Can some summary information be provided about the SES and general home living conditions of the admitted TB patients.? Discharging patients to such environments could be counterproductive, especially if patients are non-compliant with taking meds (and potentially spreading TB to the community).   

2) How was non-convergence (if any) handled for the negative binomial models?

Author Response

Dear Reviewer,

Thank you very much for the comprehensive review and constructive feedback. The detailed answers and clarifications to each of the comments are provided below.

1) Perhaps longer LOS is partly driven by hospital-acquired infections? Can some data be provided to discount this concern?

We agree that hospital-acquired infections may contribute to the overall LOS. Unfortunately, the analysis of the association between hospital-acquired infections and LOS in urogenital tuberculosis (UGTB) patients was outside the scope of this study. We did not collect data on the presence of hospital-acquired infections in the study sample. UGTB patients were not routinely screened for hospital-acquired infections at the research site and neither were these infections monitored. Based on your comment we added a limitation under the Discussion (lines 332-358):

Previous research has shown that hospital-acquired infections, particularly infections caused by drug-resistant microorganisms are common in TB patients, which prolong the length of stay. [41, 42]. Study participants were not routinely screened for hospital-acquired infections and neither were these infections monitored at RSSPMCTP. Therefore, we were not able to assess the impact of hospital-acquired infections on length of stay.

2) Can some summary information be provided about the SES and general home living conditions of the admitted TB patients? Discharging patients to such environments could be counterproductive, especially if patients are non-compliant with taking meds (and potentially spreading TB to the community).   

We mentioned in the limitations that health records of TB patients in Uzbekistan have limited information on social determinants of health. Information on patients’ education and income is not collected. There was a piece of information on living conditions recorded as a binary variable (“satisfactory” / “not satisfactory”) and all study participants had satisfactory living conditions in their records. However, the measure was not well-defined; the definition of “satisfactory” living conditions may vary across TB physicians as it was very subjective. We elaborated on the relationship between the socio-economic status and length of stay in the Discussion (lines 303-308, 329-331).

High rates of hospitalization and long length of stay might be influenced also by the low socio-economic status of TB patients. Unemployment, poverty, and homelessness are considered by health workers as social reasons to justify hospital admissions [35]. Some patients may prefer to stay in the hospital to reduce expenditure, such as nutrition or travel expenses for medication refills and treatment monitoring [36]. Previous studies in Uzbekistan have shown that one in four TB patients is unemployed, and they are at increased risk of loss to follow up [37, 38].

Patient records had limited information on social determinants of health. Health records included information on living conditions but data on socio-economic status, homelessness, and substance abuse were not available. These variables are important predictors of hospital admissions and length of stay as per published literature [40].”

2) How was non-convergence (if any) handled for the negative binomial models?

You are right, negative binomial regressions have a somewhat difficult likelihood function to maximize and therefore may result in non-convergence. In the analysis plan, we suggested exploring reasons of non-convergence and running other maximization algorithms – other than Newton-Raphson, a default option. Though, we did not have any non-convergence issues in this study.

Thanks,

Yuliia Sereda, on behalf of co-authors

Reviewer 2 Report

Authors examined the duration and determinants of hospitalizations among

patients (≥18 years) with urogenital TB (UGTB) treated with first-line anti-TB drugs during2016-18 in Tashkent, Uzbekistan. Authors showed that 96% of patients were hospitalized during the intensive phase, and that median length of stay during treatment was 56 days. The manuscript was well written, and discussion is informative. But, some parts of the manuscript need change.

  1. One of the main information of the study is hepatitis B is a significant risk factor for longer hospital stay. But, number of patients with hepatitis B is limited to be only 4. It seems to be too small to drawn conclusion. In addition, authors should describe why is stay is longer in hepatitis B patients. Is it directly related to complication of hepatitis or unfavorable response to anti-TB treatment?

  1. In this study, 34% pf patients underwent surgery for UGTB. Authors should describe what kinds of surgery they needed.

  1. In the last paragraph in discussion, authors describe that one study showed that up to 40% of hospitalizations in TB patients were avoidable [19]. Such data will help the NTP in Uzbekistan to infer hospitalization data more appropriately.

Authors had better describe definition of avoidable hospitalization in Ref 19. And how many % of patients enrolled in the present study meets the definition of Ref 19.

Author Response

Dear Reviewer,

Thank you very much for the comprehensive review and constructive feedback. The detailed answers and clarifications to each of the comments are provided below.

1. One of the main information of the study is hepatitis B is a significant risk factor for longer hospital stay. But, number of patients with hepatitis B is limited to be only 4. It seems to be too small to drawn conclusion. In addition, authors should describe why is stay is longer in hepatitis B patients. Is it directly related to complication of hepatitis or unfavorable response to anti-TB treatment?

We completely agree that the subset of patients with hepatitis B was too small and acknowledged it as a limitation (lines 360-361).

The finding on the link between the hepatitis B and longer length of stay did not have adequate statistical power given only four patients with hepatitis B in the sample.”

In this sample, prolonged hospitalization was related to complications of hepatitis B, particularly hepatotoxicity induced by anti-tuberculosis treatment. We provided details on treatment complications and final outcomes in patients with hepatitis B in the Discussion (lines 281-287).

Hepatitis B and underlying chronic liver disease are  well-known risk factors of hepatotoxicity induced by anti-tuberculosis treatment and poor outcomes [29, 30]. In our sample, all four participants with hepatitis B developed hepatotoxicity during intensive phase of TB treatment, and three of them had prolonged initial hospitalization  with about three months duration. Unfavorable treatment outcome – treatment failure - was observed in one of four TB patients with hepatitis B, which had extended intensive phase hospitalization and repeated hospitalizations during the continuation phase.”

2. In this study, 34% of patients underwent surgery for UGTB. Authors should describe what kinds of surgery they needed.

Unfortunately, we did not collect data on the frequencies of specific surgeries. To mitigate this limitation, we mentioned the common surgeries performed in UGTB patients in Uzbekistan (lines 292-294). These were nephroureterectomy, nephrectomy, urinary diversion, orchiectomy, ureteral stent placement, and reconstructive urethral surgery.    

3. In the last paragraph in discussion, authors describe that one study showed that up to 40% of hospitalizations in TB patients were avoidable [19]. Such data will help the NTP in Uzbekistan to infer hospitalization data more appropriately. Authors had better describe definition of avoidable hospitalization in Ref 19. And how many % of patients enrolled in the present study meets the definition of Ref 19.

Thank you for the comment. Based on your comment we added description of the criteria for avoidable hospitalizations in TB patients from reference 19 and another study (lines 364-374).

One study in the United States showed that up to 40% of hospitalizations in TB patients were avoidable [19]. This study considered both clinical and social criteria for hospitalization: clinically unstable on admission; admitted to the intensive care unit or intubated; receiving home or nursing home care; homeless or living in a congregate setting; recent alcohol/drug abuser; aged <5 years; admitting diagnosis of a severe form of TB; and presence of mental illness [19]. A study in Russian Federation found that about 20% of hospital admissions in TB patients would not be justified when clinical (severity of dis-ease), public health (risk of transmission), social (risk behaviours and socio-economic status), and health-system factors (access to outpatient care) are considered as hospitalization criteria [35]. Similar research, particularly in UGTB patients, will help the NTP in Uzbekistan to infer hospitalization data more appropriately.”

It is not possible to estimate the proportion of patients with unjustified hospitalizations in the current study given it was not defined as an objective and additional data collection is required. We emphasized that the high hospitalization rate and long length of stay that were found in our study should encourage the National TB programme to further explore hospitalization patterns and their criteria, particularly the proportion of avoidable hospitalizations, in pulmonary and extrapulmonary TB patients in Uzbekistan.

Thanks,

Yuliia Sereda, on behalf of co-authors

Round 2

Reviewer 2 Report

This article became better after the revision.